# A Systematic Review and Meta-Analysis of Malignant Rhabdoid and Small Cell Undifferentiated Liver Tumors: A Rational for a Uniform Classification

**DOI:** 10.3390/cancers14020272

**Published:** 2022-01-06

**Authors:** Juri Fuchs, Anastasia Murtha-Lemekhova, Markus Kessler, Fabian Ruping, Patrick Günther, Alexander Fichtner, Dominik Sturm, Katrin Hoffmann

**Affiliations:** 1Department of General, Visceral and Transplantation Surgery, University Hospital Heidelberg, 69120 Heidelberg, Germany; juri.fuchs@med.uni-heidelberg.de (J.F.); anastasia.lemekhova@med.uni-heidelberg.de (A.M.-L.); 2Generating Evidence for Diagnosis and Therapy of RarE LIVEr Disease: The RELIVE Initiative for Systematic Reviews and Meta-Analyses, University Hospital Heidelberg, 69120 Heidelberg, Germany; markus.kessler@med.uni-heidelberg.de (M.K.); patrick.guenther@med.uni-heidelberg.de (P.G.); alexander.fichtner@med.uni-heidelberg.de (A.F.); 3Department of General, Visceral and Transplantation Surgery, Division of Pediatric Surgery, University Hospital Heidelberg, 69120 Heidelberg, Germany; fabian.ruping@med.uni-heidelberg.de; 4Department of Pediatrics I, Division of Pediatric Gastroenterology, University Children’s Hospital Heidelberg, 69120 Heidelberg, Germany; 5Department of Pediatric Hematology and Oncology, Heidelberg University Hospital, 69120 Heidelberg, Germany; dominik.sturm@med.uni-heidelberg.de; 6Hopp Children’s Cancer Center (KiTZ), 69120 Heidelberg, Germany

**Keywords:** malignant rhabdoid tumor, hepatoblastoma, small cell undifferentiated (SCUD) hepatoblastoma, *SMARCB1*, INI1, pediatric liver tumors

## Abstract

**Simple Summary:**

Malignant rhabdoid tumors of the liver are very rare pediatric liver tumors with a devastating prognosis. It is currently unclear which histological subtypes of pediatric liver tumors belong to this entity and how these tumors should be treated. In this systematic review with meta-analysis, we analyzed all reports on pediatric patients with malignant rhabdoid liver tumors, but also with so-called small cell undifferentiated liver tumors. This is another rare liver tumor subtype that has recently been regarded to belong to the entity of rhabdoid tumors by some authors. The main result of this study is that these two tumor subtypes show large overlap on several levels and even mixtures of both histological patterns have been documented. Our meta-analysis provides an evidence base for the recommendation to classify these two tumor subtypes as one entity. We showed that treatment of these tumors with hepatoblastoma directed chemotherapy is ineffective and that a therapy with chemotherapy regimens initially applied for soft tissue sarcoma is associated with a significantly better survival. This study represents the highest level of evidence available for these rare liver tumors.

**Abstract:**

**Background:** Rhabdoid liver tumors in children are rare and have a devastating prognosis. Reliable diagnosis and targeted treatment approaches are urgently needed. Immunohistochemical and genetic studies suggest that tumors formerly classified as small cell undifferentiated hepatoblastoma (SCUD) belong to the entity of malignant rhabdoid tumors of the liver (MRTL), in contrast to hepatoblastomas with focal small cell histology (F-SCHB). This may have relevant implications on therapeutic approaches. However, studies with larger cohorts investigating the clinical relevance of the histological and genetic similarities for patients are lacking. **Purpose:** To analyze possible similarities and differences in patient characteristics, tumor biology, response to treatment, and clinical course of patients with MRTL, SCUD and F-SCHB. Applied therapeutic regimens and prognostic factors are investigated. **Methods:** A systematic literature search of MEDLINE, Web of Science, and CENTRAL was performed for this PRISMA-compliant systematic review. All studies of patients with MRTL, SCUD and F-SCHB that provided individual patient data were included. Demographic, histological, and clinical characteristics of the three subgroups were compared. Overall survival (OS) was estimated with the Kaplan–Meier method and prognostic factors investigated in a multivariable Cox regression model. **Protocol registered:** PROSPERO 2021 CRD42021258760. **Results:** Fifty-six studies with a total of 118 patients were included. The two subgroups MRTL and SCUD did not differ significantly in baseline patient characteristics. However, heterogenous diagnostic and therapeutic algorithms were applied. Large histological and clinical overlap between SCUD and MRTL could be shown. Two-year OS was 22% for MRTL and 13% for SCUD, while it was significantly better in F-SCHD (86%). Chemotherapeutic regimens for hepatoblastoma proved to be ineffective for both SCUD and MRTL, but successful in F-SCHB. Soft tissue sarcoma chemotherapy was associated with significantly better survival for MRTL and SCUD, but was rarely applied in SCUD. Patients who did not undergo surgical tumor resection had a significantly higher risk of death. **Conclusions:** While F-SCHB is a subtype of HB, SCUD should be classified and treated as a type of MRTL. Surgical tumor resection in combination with intensive, multi-agent chemotherapy is the only chance for cure of these tumors. Targeted therapies are highly needed to improve prognosis. Currently, aggressive regimens including soft tissue sarcoma chemotherapy, extensive resection, radiotherapy or even liver transplantation are the only option for affected children.

## 1. Introduction

Article titles like “Small Cell Undifferentiated (SCUD) Hepatoblastomas: All Malignant Rhabdoid Tumors?” [1] or “Malignant Rhabdoid Tumor, an Aggressive Tumor Often Misclassified as Small Cell Variant of Hepatoblastoma” [2] highlight a recent and pressing question in the research field of pediatric liver tumors: should malignant rhabdoid liver tumors (MRTL) and small cell undifferentiated hepatoblastomas be classified and treated as two different entities, or do they share vital features that justify merging into one entity?

Both tumors are rare primary liver neoplasms in children and show highly aggressive behavior. MRTL account for about 3% of all primary liver malignancies of childhood [1,2]. First described as a distinct liver tumor by Gonzalez-Crussi et al. in 1982 [3], the prognosis of affected children remains poor [4,5]. Historically, the diagnosis of MRTL was based on typical histological appearance of the tumor cells, with a so-called rhabdoid morphology. Genetic analyses of MRTL cells revealed typical mutations of the *SMARCB1* gene on chromosome band 22q11.2. This causes loss of function of a chromatin remodeling complex, which acts as an important tumor suppressor. By immunohistochemistry, mutations in this gene can be detected as a loss of INI1 protein expression [2]. In contrast, hepatoblastomas (HB) are typically positive for INI1.

In contrast to the devastating survival rates of MRTL, the prognosis of pediatric patients with HB has dramatically improved within the last 40 years, and long-term OS of over 80% has been achieved [6]. However, a certain group of patients with tumors showing small cell histology is still associated with poor prognosis: Historically subsumed under the entity of “anaplastic HB”, the term small cell undifferentiated HB (SCUD) was later introduced for this specific tumor. This term refers to the histological appearance of the predominant cells [7]. In historical studies, SCUD accounted for about 2% of all pediatric liver tumors classified as HB [8,9].

Recently, expert liver pathologists re-analyzed small patient cohorts or tissue biobanks and showed that SCUD share the important loss of INI1 with MRTL [1,2,10,11]. Furthermore, they found a vast predominance of small undifferentiated cells and absence of typical HB histology. Thus, some authors suggested that SCUD are in fact MRTL of the liver with a non-rhabdoid morphology. Based on these findings, they proposed to abandon the classification of SCUD as a HB subtype [1,2]. The question of whether SCUD belongs to the entity of MRTL is clinically highly relevant, as tumors classified as SCUD are still treated under regimens conceptualized for HB with inadequately low response rates. In contrast, progress in the treatment of MRTL has been made by applying aggressive treatment regimens different from those for HB, including chemotherapy regimens initially targeted against soft tissue sarcoma, radiation, and extensive surgery.

Recently, most experts came to the opinion that tumors formerly classified as SCUD should be regarded as subtype of MRTL, when INI1-staining is negative [12]. However, in the current Children’s Oncology Group (COG) classification of pediatric liver tumors [13] as well as in the most recent one by the American College of Pathologists [14], SCUD are continuously classified as a HB subtype. While new case reports have shown that not only MRTL, but also most SCUD have a loss of INI1, the clinical relevance and consequences have not been investigated yet. The current unclear situation leads to unstandardized diagnoses and therapies of these tumors. For example, some authors classify tumors with non-rhabdoid morphology but negative INI1-staining as MRTL, whereas others still adhere to the classification as SCUD [5,10,13,14]. Overall, there is an evident lack of clinical data on both MRTL and SCUD.

In this context, HB with focal small cell histology must also be addressed (F-SCHB). Since the methods of histological analysis have improved over the years, it was found that a subset of predominantly typical HB contain areas or nests with small cell histology. Moreover, for those F-SCHB where testing for INI1-expression is available, INI1 positivity even in the small tumor cells was documented [15]. Depending on whether neoadjuvant chemotherapy is applied, or upfront resection is performed, the percentage of F-SCHB differs between 1% and 12% [1,9]. This difference can be explained by the generally favorable response of HB to chemotherapy, resulting in partial tumor necrosis. This in turn may hide potential nests of small tumor cells in the specimen at the time of resection. Clinically, patients with F-SCHB usually respond to standard HB chemotherapy and tend to have a substantially better prognosis than those with MRTL or SCUD [9,15]. First evidence suggests that F-SCHB are a distinct subgroup of HB and do not belong to the entity of MRTL or SCUD. However, clinical data comparing these tumors is completely lacking.

In this systematic review with meta-analysis, we summarize all reports on pediatric patients with MRTL, SCUD and F-SCHB, analyze and compare patient characteristics, immunohistological features, applied therapies, and outcomes of these subgroups. The main research question is whether MRTL and SCUD should be classified and treated as one entity. In addition, prognostic factors are investigated, including analyses of the applied chemotherapy, surgery, and radiotherapy.

## 2. Materials and Methods

### 2.1. Review Structure and Search Strategy

This systematic review (SR) is based on a structured methodology that had been conceptualized previously by the authors, with the specific aim of generating evidence for rare diseases. It is part of the *RELIVE* research initiative (Generating evidence for RarE LIVEr Disease). The methods are in accordance with the *Preferred Reporting Items for Systematic Reviews and Meta-Analyses for Individual Patient Data (PRISMA-IPD)* guidelines. Before starting the analysis, the SR was registered with the International *Prospective Register of Systematic Review* (PROSPERO 2021 CRD42021258760).

The following free text and medical subject heading (MeSH) terms were used for a systematic literature search of three different databases (MEDLINE via PubMed, Web of Science, and CENTRAL): Rhabdoid Tumor, rhabdoid, small cell undifferentiated, small cell variant, small cell tumor, SCU, SCUD, liver, hepatic, hepatoblastoma, liver neoplasms. The detailed search algorithm is provided in the Appendix A. Moreover, references of the relevant studies were screened for eligible articles. The last search was performed on 11th October 2021.

### 2.2. Study Selection Criteria and Selection Process

All study types were eligible for this systematic review. The following inclusion criteria were applied:

Patient age < 18 years;

Primary liver tumor;

Histological diagnosis of malignant rhabdoid tumor or small cell undifferentiated hepatoblastoma or focal small cell histology in hepatoblastoma;

Individual patient data on age, therapy, and outcome available.

Historic reports of patients with, at the time, so-called “anaplastic hepatoblastoma” were only included if the information of histological tumor characteristics allowed for a reliable classification as SCUD. Moreover, studies with only aggregated data accessible, and no report of subgroup outcomes of patients with MRTL, SCUD or F-SCHB, were not suitable for this analysis and thus excluded.

The list of studies retrieved by the systematic search was screened for eligible studies by two reviewers independently (JF and AML). After a first selection based on the abstracts, the two reviewers worked through the full texts of all eligible studies and decision on in- or exclusion was made. Dissent between the two reviewers was resolved after consulting with a third reviewer (KH).

### 2.3. Data Extraction and Investigated Variables

Data was extracted by means of a standardized form, that had been validated with data extraction of the first five studies. The collected data items were compiled after reviewing the relevant literature on pediatric liver tumors. Independently from each other, the two reviewers extracted the data based on the predefined form.

### 2.4. Risk of Bias Assessment

Given the rarity of the investigated tumors, no randomized trials were expected to be found. For observational studies, the validated MINORS tool was applied for the risk of bias (RoB) assessment [16]. As case reports/series were included in our analysis, a RoB tool specifically designed for this methodology was used [17].

### 2.5. Statistical Analyses and Certainty of Evidence

R (version 3.6.2, Vienna, Austria) [18] was used for all statistical analyses, with the the survminer [19] and survival [20] packages for survival curves and the forestmodel [21] package for forest plots. Patient data were entered individually into a database. Patient data were pooled, and patients were divided into three subgroups according to diagnosis made in the studies: MRTL, SCUD, or F-SCHB. For descriptive statistics of continuous data, means, medians, standard deviations (SD) or interquartile ranges (IQR) were calculated. For categorial data, numbers with percentages are given. Chi-squared tests (without Yate’s correction) or Mann–Whitney U tests were used for univariate analyses, at a level of significance of 5%. Overall survival (OS) and progression free survival (PFS) were calculated with the Kaplan–Meier method. Univariate significance of variables for OS/PFS was tested with the log rank test (level of significance 5%). Factors that showed univariate significance were included in multivariable cox regression models for calculating the independent hazard ratios (HR) of predictive variables.

The GRADE criteria were applied for determining the certainty of evidence and strength of recommendations [22].

### 2.6. Applied Terms and Definitions

*Staging*. The widely applied and validated PRETEXT system for pediatric liver tumors was used for staging the local extent of disease [23].

*Histology.* Liver tumors were considered to have rhabdoid morphology when large, polygonal tumor cells with pronounced eosinophilic cytoplasm, and large eccentric nuclei with centrally visible nucleoli were found. Intracellular inclusions with positivity for vimentin or other intermediate filament proteins are typically found in a subset of rhabdoid tumor cells. The term small cell undifferentiated was used for liver tumors with round or oval cells with sparse cytoplasm, faint nuclei with unremarkable nucleoli. These cells may express vimentin or different types of cytokeratin and are negative for alpha-feto protein. F-SCHB was defined as liver tumor with focal small cell histology mixed with other hepatoblastoma-typical tumor cells, and less than 50% of areas with small undifferentiated cells in the specimen.

*Surgery*. Upfront surgery was defined as operation with intent of primary tumor resection before starting a chemotherapy. R0 was defined as microscopically tumor-free resection margins. R1 as macroscopic tumor resection but microscopically positive margins. R2 was defined as macroscopic tumor residual.

*Chemotherapy*. Chemotherapy with neoadjuvant intent was defined as treatment before a planned surgical tumor resection. In many cases, the intended surgery was not performed due to rapid disease progression during neoadjuvant chemotherapy. The term *Hepatoblastoma chemotherapy* (HB CT) was defined as regimens used in HB trials or protocols. Soft tissue sarcoma chemotherapy (STS CT) was defined as regimens that had been first conceptualized or applied in trials or protocols for soft tissue sarcoma. No response was defined as no reduction or increase in tumor size during chemotherapy. Partial response was defined as reduction of tumor size of 20–50%. Good response was defined as reduction of tumor size >50% and/or disappearance of synchronous metastatic lesions on imaging.

*Outcome.* Disease related death (DRD) was defined as death of a patient either caused by progressive disease and consecutive complications or by complications of the treatment (e.g., postoperative death, toxicity of chemotherapy). Complete remission was defined as no evidence of disease after the end of treatment. Partial remission was defined as reduction of tumor size that changed the staging (e.g., change of unresectable to resectable, complete resection of primary tumor with persisting distant disease). Overall survival (OS) was defined as time from diagnosis to the last follow-up or death of a patient. Progression free survival (PFS) was defined as time from diagnosis to increase of tumor size (primary or metastatic), occurrence of new metastatic lesions or serious complications directly associated with the disease, or treatment side effects.

## 3. Results

### 3.1. Literature Search and Study Selection

The study selection process is depicted in the PRISMA flow diagram (Figure 1). The 444 records found by the systematic literature search were screened for eligibility, and another 13 studies were found by going through the reference lists of relevant studies. Fifty-six studies were eventually included in our analysis, of which seven studies were retrospective observational studies and 49 case reports or series. These studies provided information for 118 patients, of which 55 were diagnosed with MRTL, 41 with SCUD and 22 with F-SCHB. The full list of all included studies is provided in the Appendix A.

### 3.2. Critical Appraisal of Included Studies and Risk of Bias

In light of the rarity of the investigated liver tumors, conduct of prospective or large studies is challenging, demands extensive organization and international coordination. Therefore, no randomized controlled trials and no prospective studies that specifically investigated MRTL, SCUD or F-SCHB have been conducted yet. Moreover, no study was found that investigated clinical similarities or differences of patients with MRTL and SCUD.

The certainty concerning the diagnosis of either MRTL, SCUD or F-SCHB was high throughout all included studies. For the majority of the patients, inclusion in an oncologic trial registry was explicitly reported (55%). This implied that the therapeutic regimens were administered after consultation with a coordinating study center or patients received predefined therapies according to a study protocol. Nevertheless, a large variety of therapeutic approaches was found in the included studies, mirroring a lack of standardized interventions for MRTL and SCUD. In contrast, all patients with F-SCHB received standard regimens according to HB trials. Most of the included studies are case reports or small series of pediatric patients. Case reports inevitably introduced selection bias. However, the high percentage of reports with fatal outcome suggests that reporting bias for positive results was limited. The quality of the case reports differed substantially. Five case series had a low RoB. Detailed description of cases with complete data on patient characteristics, therapy and outcome was provided. Most case reports had a moderate RoB (27/49), with high selection bias, but detailed data on patient characteristics and outcome, with varying quality concerning information of the applied therapies as well as follow-up data. Seven studies were retrospective observational studies. One of them was a subgroup analysis of patients treated in the SIOPEL studies 1, 2 and 3 [24] with a low overall risk of bias (13/16 points in MINORS). Despite the low number of patients (six cases), the studies by Cornet et al. and Fazlollahi et al. provided interesting insights with detailed and comprehensive data on staging, histology, applied therapy and outcome [2,5]. The studies by Haas et al. and Zhou et al. were important concerning the differentiation of F-SCHB from SCUD and MRTL and had an acceptable RoB [9,15]. The studies by Lautz et al. and Bajpai et al. had not been specifically designed to investigate MRTL or SCUD; however, they provided adequate data for individual patients with MRTL/SCUD within their series [25,26]. Overall, most studies provided detailed descriptions of cases and an acceptable to good level of data integrity. However, a lack of studies with higher caseloads is obvious, as well as the absence of studies investigating the best suited classification and treatment of SCUD and MRTL. The tables with RoB for all included studies are provided in the Appendix A. The Prisma IPD-checklist can be found in the Appendix A. 

### 3.3. Composition of Subgroups and Patient Characteristics

Of the 118 patients included in this analysis, 55 were diagnosed with MRTL, 41 with SCUD and 22 with F-SCHB. The median age at diagnosis did not differ significantly between patients with MRTL and those with SCUD (7 vs. 10 months, z-score −1.86, *p* = 0.061). The median age of patients with F-SCHB was significantly higher than this of patients with MRTL and SCUD (14 months, z-score −1.97, *p* = 0.049). In all three subgroups, there was a predominance of male gender. Initial misdiagnosis, mostly as hepatoblastoma, was frequently reported for patients who were diagnosed in the final pathology report with MRTL (20 cases, 36%). Elevated alpha-feto protein (AFP) was typical for patients with F-SCHB (73%) in contrast to patients with MRTL and SCUD (9% and 5%).

The vast majority of MRTL and SCUD patients presented with large and locally advanced tumors (≥PRETEXT III in 71% and 78%, respectively). Salient is the high number of patients with tumor rupture (before start of any treatment) in the MRTL subgroup (9 cases, 16%). An overview of the patients characteristics is given in Table 1. 

### 3.4. Differentiation of MRTL/SCUD from F-SCHB

While the analysis showed large overlap concerning patient characteristics, tumor biology, and outcomes between MRTL and SCUD (see the following paragraphs), several results showed clear distinction of F-SCHB from the other two. The median age of 14 months of patients with F-SCHB in our analysis is within the range of the typical age of onset for hepatoblastoma that is described in the literature [27]. Patients with F-SCHB had significantly less often metastasis at diagnosis compared to those with MRTL and SCUD (MRTL vs. F-SCHB: OR 14.0, 95% CI 1.8–303.6, *p* = 0.002; SCUD vs. F-SCHB: OR 13.4, 95% CI 1.6–298.8, *p* = 0.003). In all F-SCHB with INI1 status available, INI1-staining was positive for all tumor cells. In contrast, 100% of MRTL and 92% of SCUD with available data on INI1 status were negative for INI1. While almost all F-SCHB showed response to standard HB chemotherapy, most patients with MRTL and SCUD had disease progression during chemotherapy (Response rate of MRTL vs. F-SCHB: 27% vs. 86%, OR 0.1, 95% CI 0.0–0.3, *p* < 0.001; SCUD vs. F-SCHB: 24% vs. 86%, OR 0.1, 95% CI 0.0–0.24, *p* < 0.001). The survival rate of patients with F-SCHB differed significantly from the outcome of patients with MRTL and SCUD. The mortality was 14% in the F-SCHB subgroup, versus 76% in the MRTL, and 93% in the SCUD subgroup (MRTL vs. F-SCHB: OR 20.5, 95% CI 4.6–104.5, *p* < 0.001; SCUD vs. F-SCHB: OR 80.2, 95% CI 12.1–709.5, *p* < 0.001, see Table 2 for details). Two-year OS was significantly higher for F-SCHB compared to MRTL and SCUD (F-SCHB: 86%, MRTL: 22%, SCUD: 13%, *p* < 0.001, Figure 2). Given these significant differences between F-SCHB on one side, and MRTL/SCUD on the other, F-SCHB was regarded as a subtype of hepatoblastoma and as a distinct entity different from MRTL/SCUD in our study. Consequently, the analysis focused on the investigation of possible similarities and differences of MRTL and SCUD, with the main question of whether these tumors should be classified as one entity.

### 3.5. Comparison of MRTL and SCUD

*Baseline patient data*. The median age at diagnosis did not differ significantly between the two subgroups of MRTL and SCUD (z-score = −1.87, *p* = 0.061). The rates of male gender, non-elevated AFP, PRETEXT III or IV tumors, multifocal tumors, and metastasis at diagnosis were all similar among the two subgroups, without any significant differences (See Figure 3 for details).

*Histology.* The vast majority of tumor diagnosed as MRTL had rhabdoid morphology. However, eight patients (15%) were diagnosed with MRTL despite the absence of typical rhabdoid tumor cells. Interestingly, there were four tumors with rhabdoid cells that were classified as SCUD. All MRTL with available INI1 status were negative. Similarly, all but one SCUD with available INI1 staining were negative. Important to note is that tumors with a mixture of both, typical rhabdoid and small undifferentiated cells were reported.

*Classification as MRTL or SCUD*. The diagnostic criteria for MRTL and SCUD differed across the included studies and case reports. In studies conducted before the identification of *SMARCB1* mutations with consecutive loss of INI1-positivity in immunohistochemistry, the classification of liver tumors as MRTL were solely based on cytomorphology (i.e., predominance of rhabdoid tumor cells). With INI1 staining available and the discovery of loss of INI1 in liver tumors without rhabdoid morphology, some centers and authors changed their classification algorithm. In recent studies, some authors defined MRTL as liver tumors with loss of INI1 independent of the presence of rhabdoid or small undifferentiated tumor cells [2,5]. Others still adhered to the classification of small cell undifferentiated HB for tumors without rhabdoid tumor cells, but INI1-negativity of undifferentiated cells [10,28,29].

*Chemotherapy regimens.* Many different chemotherapy regimens were applied in patients with MRTL and SCUD. Regimens that had been conceptualized, tested, or applied mostly for soft tissue sarcoma were significantly more often applied in patients with tumors diagnosed as MRTL compared to those with SCUD (20 vs. 4, OR 5.3, 95% CI 1.3–25.3, *p* = 0.007, Figure 2). Most patients diagnosed with SCUD received hepatoblastoma-targeted chemotherapy (26 patients, 65%. PLADO in 21 patients, 53%). A variety of different regimen were applied in both subgroups and many patients were treated with unstandardized combinations of chemotherapeutic agents. See Table 3, Figure 4 and Figure 5.

*Surgery.* Upfront surgery was performed in 18% and 20% of the patient with MRTL and SCUD, respectively. Overall, 29% (MRTL) and 27% (SCUD) underwent surgery after neoadjuvant chemotherapy. In addition, 53% (MRTL) and 54% (SCUD) did not undergo any kind of surgical tumor resection. In one patient with MRTL and in two patients with SCUD, total hepatectomy with liver transplantation was performed. All other surgeries were major liver resections. R0 resection was achieved in 46% of the operations in MRTL, and in 68% of the operations in SCUD patients. Patients with MRTL and SCUD that did not undergo surgical tumor resection had a significantly worse OS than those with upfront resection or surgery after neoadjuvant chemotherapy (Table 4, Figure 6 and Figure 7).

*Outcome*. Mean follow-up was 35 and 37 months for patients with MRTL and SCUD, respectively (median 24 and 18 months, SD 34 and 41). Overall, 76% succumbed to the disease in the MRTL subgroup (42 patient with DRD), and 93% died in the SCUD subgroup (38 patients with DRD). Kaplan–Meier estimates for 2-year OS were 22% for MRTL (95% CI 13–37) and 13% for SCUD (95% CI 5–31). This difference was not significant (*p* = 0.750). Two-year OS for the two groups pooled was 16% (95% CI 10–27). Disease progression during the initially therapy was observed in 56% of MRTL and 61% of SCUD patients. Relapse occurred in 11 patients in the MRTL subgroup (20%) and in 16 patients in the SCUD subgroup (39%). Mean time from (partial) remission to relapse was 5 months in the MRTL and 4 months in the SCUD subgroup. See Table 5 for details on the outcome.

*Prognostic factors.* Given the large overlap of patient characteristics, histopathologic features, and tumor behavior of MRTL and SCUD shown in the previous section of this study, the analysis of prognostic factors was performed under the hypothesis that MRTL and SCUD are subtypes of one tumor entity. The classification of tumors as either MRTL or SCUD was included in a multivariable cox regression analysis as a control variable, as well as PRETEXT staging. The following factors were significant predictors of survival in univariate analysis and thus included in the multivariable cox regression model (Figure 7): Age (≤12 months/>12 months), Treatment period (1980–1995/1996–2005/After 2005), M-status (no metastasis/metastasis present), Chemotherapy regimen (Hepatoblastoma/Soft tissue sarcoma/Other), Timing of surgery (Upfront/After neoadjuvant Chemotherapy/No surgery), R0 resection status (achieved/not achieved), Local radiotherapy (Yes/No). Age ≤ 12 months and absence of surgical tumor resection were significantly associated with a higher risk of death. Patients who were treated with STS CT had a significantly reduced risk of death (HR 0.14, 95% CI 0.05–0.40, *p* < 0.001). The classification as either MRTL or SCUD was not an independent predictor of death.

In both subgroups, all patients that survived (disease free at last follow-up) underwent surgical tumor resection with chemotherapy (mostly neoadjuvant and adjuvant). While in both subgroups (total 96 patients) only seven patients received radiotherapy (RT) (7%), three patients among the 11 disease free survivors were treated with adjuvant RT (27%).

## 4. Discussion

This is the first systematic review on MRTL and SCUD and the first study with clinical comparison of those two very rare and highly aggressive liver tumors in children. The largest number of cases with MRTL and SCUD hitherto reported in the literature was analyzed. Both tumors have a devastating prognosis with mortality rates of 76% for MRTL and 93% for SCUD. Our study demonstrates large overlap in onset and course of the disease, demographic patient characteristics, response to therapy, and outcomes between both tumor subtypes. In combination with the biological and genetic similarities between MRTL and SCUD, that have been shown previously by expert pathologists [1,2,11,30], our results support the hypothesis that SCUD belong the group of rhabdoid tumors rather than being a subtype of hepatoblastoma. However, it must be acknowledged that further genetical analyses of the two histological tumor types are necessary to gain further certainty concerning the possible classification as one entity. While some improvement in the management of MRTL has been achieved by applying chemotherapeutic regimens that are specific for soft tissue sarcoma, classical hepatoblastoma chemotherapy proved to be ineffective for both MRTL and SCUD. Moreover, surgical tumor resection is essential for preserving a chance of cure for the affected patients.

### 4.1. Importance of INI1-Status and Its Clinical Relevance

The diagnosis of MRTL is difficult, which is mirrored by more than one third of patients that were initially misdiagnosed as HB. INI1-status emerged as an important diagnostic tool to corroborate a suspected diagnosis of MRTL, which otherwise relied on cytomorphology [31]. Pathology reviews and analysis of tumor biobanks have repeatedly described the histopathological overlap of MRTL and SCUD, and in particular the loss of the INI1 also in tumors classified as SCUD [1,2,32]. These findings are confirmed by our meta-analyses. Our study showed that some centers concluded that SCUD are in fact rhabdoid tumors and have already changed classification and treatment algorithm accordingly, while others stuck to the classification as *“small cell undifferentiated hepatoblastoma”* [10]. As mentioned above, current classifications of pediatric liver tumors do not yet recommend classifying and treating SCUD as MRTL, even when loss of INI1 is demonstrated [12,13,14]. Furthermore, our results show that HB with focal small cell histology have preserved INI1 expression within the small cell areas and are associated with a significantly better prognosis than those liver tumors with loss of INI1. F-SCHB typically respond to HB chemotherapy and must be strictly distinguished from MRTL or SCUD. F-SCHB should be classified as a subtype of HB and be treated within HB trials. The clinical relevance and prognostic value of small cell areas needs to be further investigated.

### 4.2. Why SCUD Should Be Classified as MRTL

Several studies on large cohorts of HB patients confirmed the significantly worse prognosis for patients with SCUD treated in HB-trials compared to all other histological subtypes [8,33]. Similarities between SCUD and MRTL haven been mentioned for already over 10 years. In 2009, Trobaugh-Lotrario et al. presented a review of historical cases of SCUD together with selected cases from personal communication [34]. The authors did not separately analyze F-SCHB and SCUD, however, they suggested that at least a subset of patients with liver tumors presenting small cell undifferentiated histology, might be more similar to rhabdoid tumors than to HB. Other authors supported these findings, albeit mainly based on histological investigations rather than analyses of clinical and therapeutical relevance [1,11,30]. For rhabdoid tumors of the central nervous system, it has long been recognized that rhabdoid morphology is only present in a subset of all rhabdoid tumors and small cell types are also encountered [35]. However, the classifications of pediatric liver tumors has not been changed concerning SCUD so far [13]. Our study demonstrated not only (immuno)histological, but also large clinical overlap between MRTL and SCUD. Several insights of our analyses substantiate the hypothesis that MRTL and SCUD belong to the same tumor entity and that SCUD may not be treated as a subtype of HB. Detailed histological examinations of tumors showed that mixed forms with typical rhabdoid morphology and small undifferentiated cells exist within one tumor [1,36]. We showed that the two patient groups do not differ significantly in relevant categories, such as patient age, elevation of AFP, INI1 status, and tumor stage at diagnosis. Significant differences were only found in the applied treatments: most patients with tumors classified as SCUD were treated with HB chemotherapy, while MRTL has increasingly been treated with STS CT in recent years. It is important to note that some tumors without rhabdoid morphology, but loss of INI1 were classified and treated as MRTL by some authors. At the same time, others classified and treated such tumors as SCUD [10]. These unstandardized classifications and treatments emphasize the need for recommendations concerning the best suited classification and therapy for these rare tumors. Our results clearly prove that classical HB chemotherapy is not only unsuited for MRTL, but also largely ineffective in SCUD. Less than 10% of patients with SCUD and therapy with HB chemotherapy showed treatment response. The rate of patients treated with HB chemotherapy was significantly higher in the SCUD than in the MRTL subgroup. Based on the hypothesis that HB chemotherapy is not a suited treatment in SCUD, the even worse outcome in the SCUD subgroup compared to patients with tumors classified as MRTL (mortality 93% vs. 76%) might be explained to some extent. In the MRTL subgroup, a significant higher number of patients were treated with STS CT, which proved to significantly reduce the risk of death. 50% of the MRTL patients who were treated with STS CT survived the disease. Notably, two of three patients with tumors missing typical rhabdoid morphology but showing INI1 negativity, survived disease free after therapy with STS CT. These findings strongly reinforce the recommendation of classifying and treating tumors with small cell undifferentiated histology and loss of INI1 as malignant rhabdoid liver tumors.

### 4.3. Predictive Factors and Recommendations for Therapy

*Age at diagnosis.* Early onset of the disease was associated with poorer outcome in our analyses. While this significant result is described for malignant rhabdoid tumor of the liver for the first time, younger age has already been identified as a negative predictor of survival in previous studies on rhabdoid tumors at various primary sites [4]. To some extent, this can be explained by a subset of cases with congenital or neonatal tumors in children with germline mutations of the *SMARCB1* gene and consecutive rhabdoid tumor syndrome. Moreover, aggressive therapies are less likely to be administered in neonatal or very young patients, which might further decrease the chances for survival of these children.

*PRETEXT staging.* Interestingly, PRETEXT staging was not predictive for survival of patients with MRTL and SCUD. However, given the rapid progression of these tumors, the vast majority of patients are diagnosed with large and locally advanced tumors of PRETEXT III or IV. This weakens the validity of this staging method that has been validated for HB. Only 10% of MRTL and SCUD were staged as PRETEXT I or II.

*The role of surgery.* A disease-free survival was only observed in 13 patients with MRTL or SCUD, all of which were treated with chemotherapy and surgical tumor resection. Three of them received additional radiotherapy. To date, a chance of cure for these aggressive liver neoplasms is only preserved if multimodal therapeutic regimens are applied, in which surgical tumor resection is an integral part. This finding was further emphasized by the fact that patients without any surgical tumor resection had a significantly increased risk of death in the multivariable cox regression analysis. The role of radiotherapy in the treatment of MRTL and SCUD has to be further investigated. Initial evidence presented by our study suggests that there is in fact an important role for radiation in the treatment of these tumors.

*Soft tissue sarcoma chemotherapy.* Patients with MRTL and SCUD who were treated with STS CT had a significantly reduced risk of death. The majority of the disease-free survivors were treated with STS CT. Our results show that these regimens are to date the best suited chemotherapeutic treatments available for MRTL. Our study strongly suggests that also for patients with small cell undifferentiated, INI1 negative liver tumors should be treated with these protocols. However, further studies are highly needed given the small number of these cases treated with STS CT. The most applied STS CT regimens were CEVAIE, ICE and VAC-IE. CEVAIE and VAC-IE achieved better results than ICE. However, this can only be regarded as initial evidence and further investigations of the best suited regimens are required. Within the European Paediatric Soft Tissue Sarcoma Study Group Non-Rhabdomyosarcoma Soft Tissue Sarcoma 2005 Study (EpSSG NRSTS 2005, Padua, Italy), the largest number of patients with extra-cranial rhabdoid tumors hitherto reported (100 patients) were treated according to a soft tissue sarcoma protocol, of which 43 completed the treatment protocol [4]. Three-year OS was 38% among the whole cohort, who were treated with a regimen consisting of vincristine, cyclophosphamide, doxorubicin, carboplatin, and etoposide. Surgery and radiotherapy were not performed in a standardized algorithm, which limited the possible conclusions from this study, that only provide aggregated patient data [4]. Moreover, given the very different primary tumor sites included in this study, particularities of the manifestation in certain locations, such as the liver, were not investigated. Currently, many patients with rhabdoid tumors in Europe are treated according to the protocol of the European Rhabdoid Registry (EU-RHAB). Rhabdoid liver tumors are subsumed under the category “Rhabdoid Tumors of Soft Tissue”. The standard chemotherapy regimen for these tumors in EU-RHAB consists of Doxorubicin plus Ifosfamide-Carboplatinum-Etoposide (ICE) plus Vincristine-Actinomycin D-Cyclophosphamide (VAC) [37]. Subgroup outcomes of patients with MRTL with the treatments prescribed by EU-RHAB have not been investigated or published yet.

*The role of radiotherapy.* While the importance of radiation has been shown to atypical teratoid/rhabdoid tumors of the CNS [38,39], the potential harms or benefits of radiotherapy have not been clearly demonstrated for extracranial, and in particular rhabdoid tumors of the liver. Results of comparatively large series of patients with extracranial rhabdoid tumors of various primary sites suggest that local radiotherapy improves the prognosis [40,41], albeit the results being heterogenous and optimal timing, dosage and modalities remain unclear [42]. Local radiotherapy was only applied in 7 of 96 patients with MRTL/SCUD in our analysis (7%), which limited the possible conclusions concerning the impact of radiotherapy on the outcome of children with rhabdoid tumors of the liver. Among the 11 disease free survivors, three patients received adjuvant radiotherapy (27%). Further studies are needed to elucidate the role of radiation in the treatment of these tumors.

### 4.4. Status Quo and Prospect for Therapeutic Approaches

MRTL and SCUD are very rare liver tumors in children with a grim prognosis until the present. Initial misdiagnosis and unstandardized therapies slow down progress in the treatment of these challenging cases. Based on the hypothesis that MRTL and SCUD belong to the same entity, they account for about 5% of all primary liver tumors in children. At present, no treatment protocol or trial specifically designed for MRTL or SCUD, is being conducted. The classification, diagnostic algorithm, and therapy for these tumors has not been standardized internationally.

The European Rhabdoid Registry (EU-RHAB) has been founded in 2007 with the aim of establishing a central database for all cases of rhabdoid tumors in Europe in order to develop structured diagnostic and therapeutic protocols [37]. Remarkable insights have been yielded by this initiative, that formed the basis for improvement of the prognosis of patients with rhabdoid tumors. However, oncological registries that include tumors with various primary sites run the risk of missing details and important characteristics of certain primary locations. Concerning the case of rhabdoid liver tumors, this is exemplified by the fact that the important question of whether INI1-negative SCUD should be classified and treated as MRTL has not been specifically addressed and investigated in EU-RHAB so far. Similar concerns apply to the EpSSG NRSTS 2005 study. We strongly encourage clinicians and researcher that future trials investigating MRTL should include patients with tumors formerly classified as SCUD.

While improvement in the prognosis has been achieved by multimodal therapy including STS CT or high-dose chemotherapy, extensive surgery and radiotherapy, there is still a large part of patients with rhabdoid tumors that do not respond to conventional multimodal therapy [43]. Thus, new approaches are desperately needed. The molecular biology of rhabdoid tumors is comparatively well described and understood. There is a substantial number of potential molecular targets and possible inhibitors, mostly focusing on the *SMARCB1*-related pathways, that have even been investigated in in vitro studies [43,44]. This promised dramatic improvements in the therapy and prognosis of patients suffering from rhabdoid tumors. However, breakthrough therapeutical innovations have failed to materialize so far [43]. The reasons for this shortcoming are manifold and complex, among them being the rarity of the disease and the additional constraints of conducting clinical trials in pediatric populations. Clinical data with promising results is lacking or studies did not meet efficacy endpoints and were terminated early [43,45]. For the time being, expert opinion strongly recommends treating pediatric patients with rhabdoid tumor according to existing multimodal treatments regimens, given that a part of these patient will respond well to this therapy [43]. For those with primary or secondary resistance to standard regimens, well designed studies with new agents are highly needed.

### 4.5. Certainty of Evidence and Strength of Recommendations

The certainty of evidence produced by this systematic review must be evaluated in the context of an extremely rare disease and the low level of existing evidence. All existing and included studies were non-comparative, thus limiting the evidence to a low level and the strength of recommendations based on this study is conditional. However, our study provides a comprehensive overview of all reported cases of MRTL and SCUD and thus an evidence base for several recommendations and future trials. In particular, a uniform classification of MRTL and SCUD with consecutive standardized treatment can be based on our results. Our study represents the highest level of evidence existing for these rare pediatric liver tumors.

## 5. Key Insights and Messages

⮚Results of this meta-analysis strongly suggest that SCUD should be treated with the same therapeutic strategies as MRTL;⮚The term small cell undifferentiated hepatoblastoma is misleading and should be abandoned;⮚SCUD and MRTL are associated with alarming mortality rates;⮚Hepatoblastoma regimens are unsuited for both SCUD and MRTL;⮚Soft tissue sarcoma chemotherapy is the best suited therapy to date for MRTL;⮚Surgical resection is vital to maintain a chance of cure for children with MRTL and SCUD.

## Figures and Tables

**Figure 1 cancers-14-00272-f001:**
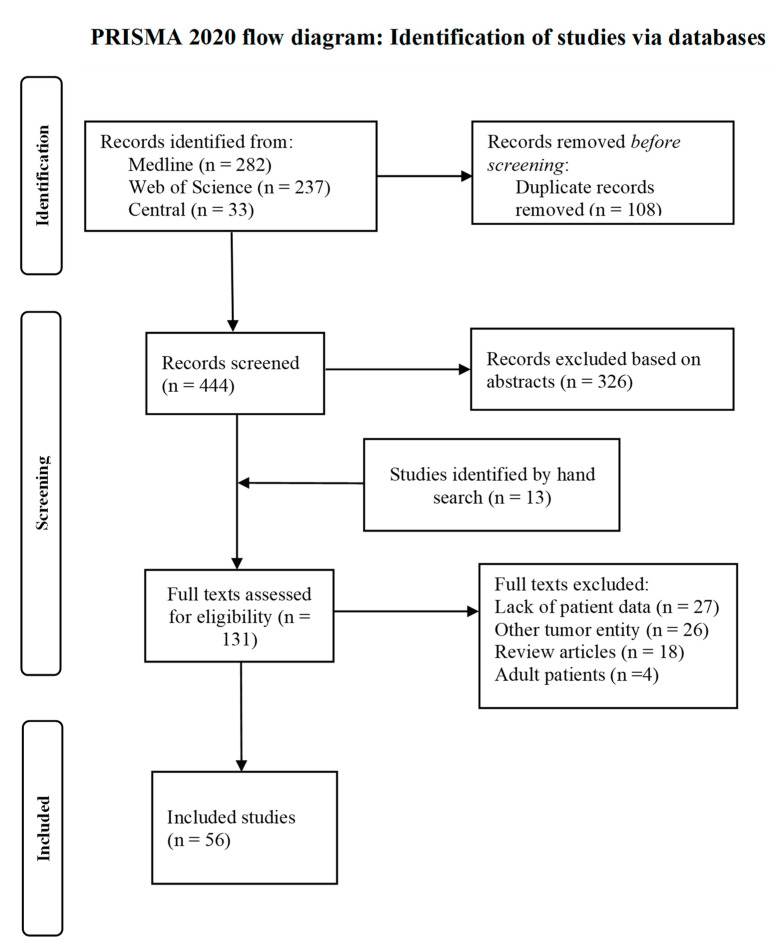
Prisma flow chart of study selection process.

**Figure 2 cancers-14-00272-f002:**
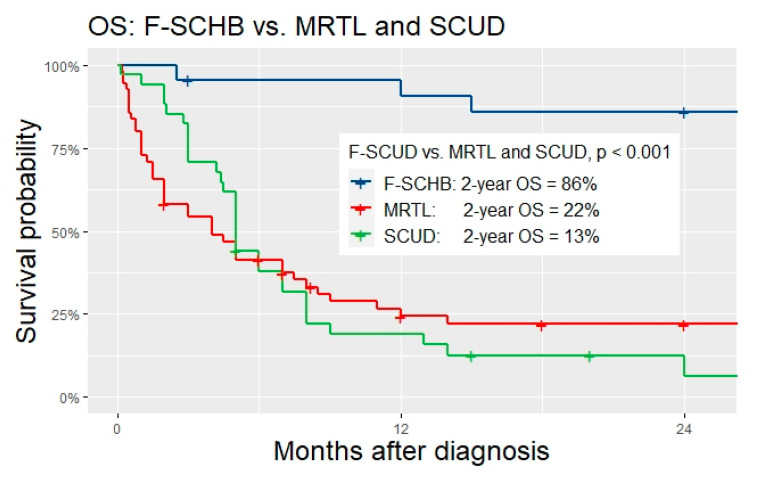
OS of patients with F-SCHD vs. MRTL vs. SCUD.

**Figure 3 cancers-14-00272-f003:**
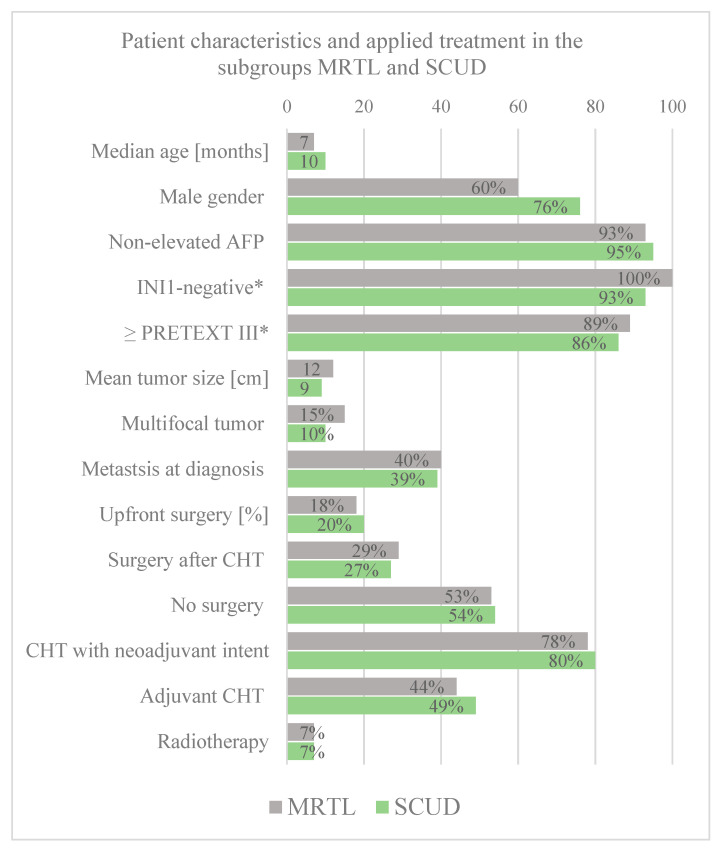
Overview of patient characteristics in the subgroup of MRTL and SCUD. * Percentages of those with available data on this variable.

**Figure 4 cancers-14-00272-f004:**
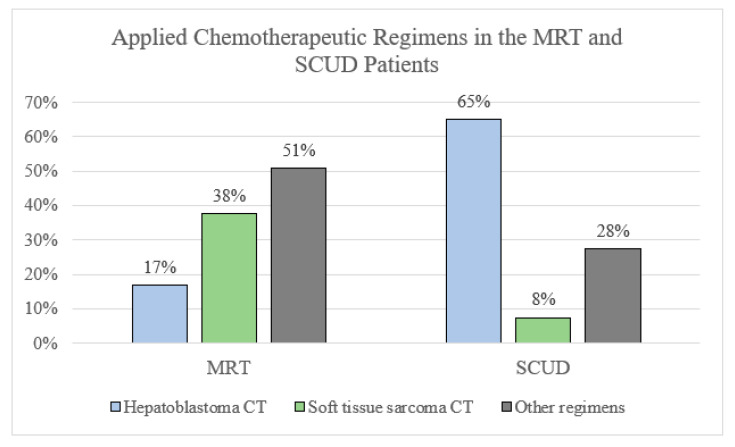
Overview of applied chemotherapy regiments in patients with MRTL vs. SCUD.

**Figure 5 cancers-14-00272-f005:**
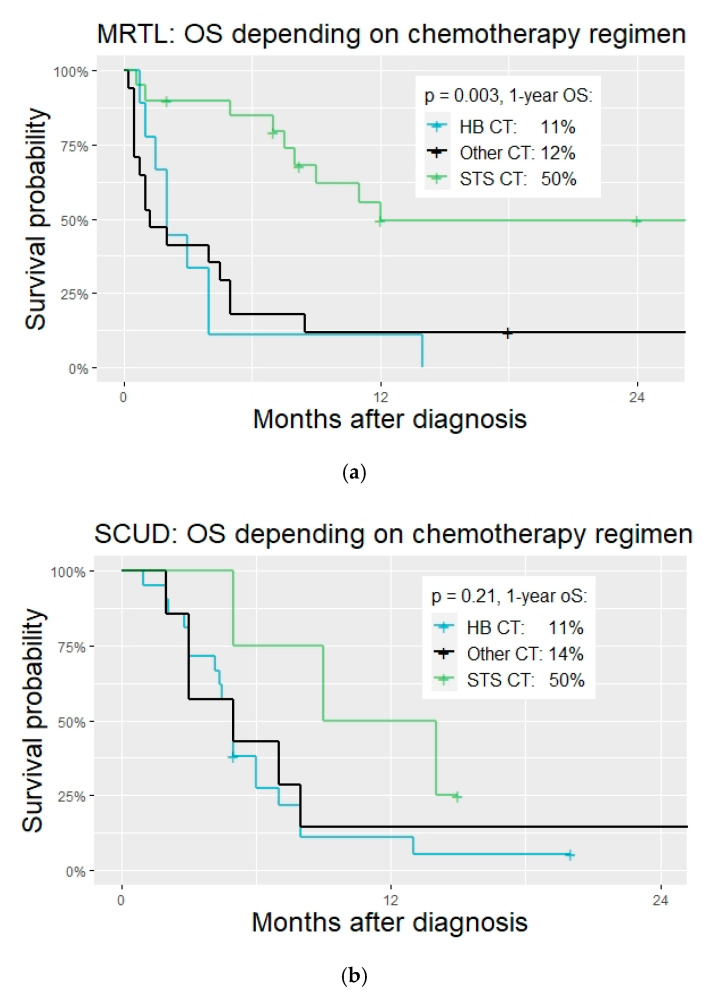
OS of patients with MRTL and SCUD according to different applied CT regimens. (**a**) OS of patients with MRTL and different CT regimens (**b**) OS of patients with SCUD and different CT regimens.

**Figure 6 cancers-14-00272-f006:**
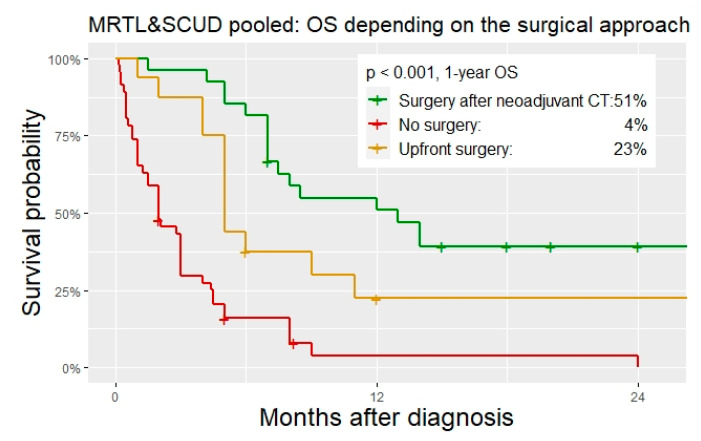
OS of the pooled group (MRTL and SCUD) for different surgical approaches.

**Figure 7 cancers-14-00272-f007:**
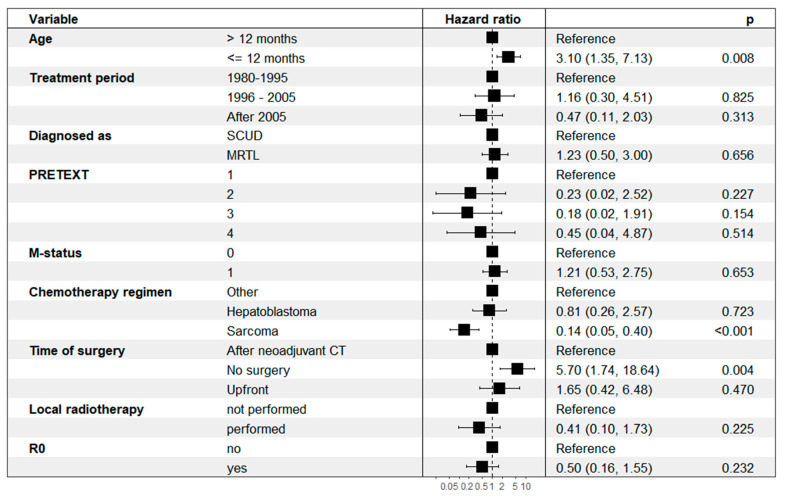
Multivariable Cox regression analysis with hazard ratios.

**Table 1 cancers-14-00272-t001:** Patient baseline characteristics.

	MRTL *n* = 55	SCUD *n* = 41	F-SCHB *n* = 22
Median age [months]	7 (IQR 4–15, range 0–180)	10 (IQR 7–24, range 0–138)	14 (IQR 6–20, range 4–48)
Congenital tumor	3 (5%)	1 (2%)	0
Gender	33 males (60%)	31 males (76%)	15 males (71%)
Comorbidity	0	2 (5%)	3 (14%)
Initial misdiagnosis	20 (36%)	3 (7%)	1 (5%)
Elevated AFP	4 (7%)	2 (5%)	16 (73%)
Histology			
Rhabdoid morphology	47 (85%)	4 (10%)	0
No rhabdoid morphology	8 (15%)	37 (90%)	22 (100%)
INI1-staining			
Negative	30 (55%)	11 (27%)	0
Positive	0	1 (2%)	7 (32%)
Not available	25 (45%)	29 (71%)	15 (68%)
Staging			
PRETEXT I	1 (2%)	0	1 (5%)
PRETEXT II	4 (7%)	5 (12%)	6 (27%)
PRETEXT III	20 (36%)	14 (34%)	0
PRETEXT IV	19 (35%)	18 (44%)	0
PREXTEXT not available	13 (24%)	4 (10%)	15 (71%)
Multifocal tumor	8 (15%)	4 (10%)	1 (5%)
Tumor rupture (before treatment)	9 (16%)	1 (2%)	0
Mean tumor size (largest diameter)	12 cm	9 cm	8 cm
Metastasis at diagnosis	22 (40%)	16 (39%)	1 (5%)
Invasion of major hepatic veins/portal vein (reported)	9 (16%)	2 (5%)	1 (5%)

**Table 2 cancers-14-00272-t002:** F-SCHB vs. MRTL/SCUD.

	Synchronous Metastasis		Response to Chemotherapy		DRD	
**F-SCHB**	**1 (5%)**		19 (86%)		3 (14%)	
MRTL	22 (40%)	**OR 14.0, *p* = 0.002**	15 (27%)	OR 0.1, *p* < 0.001	42 (76%)	OR 20.5, *p* < 0.001
SCUD	16 (39%)	** OR 13.4, *p* = 0.003**	10 (24%)	OR 0.03, *p* < 0.001	38 (93%)	OR 80.2, *p* < 0.001

**Table 3 cancers-14-00272-t003:** Applied chemotherapy regimens.

	MRTL *n* = 55	SCUD *n* = 41
Hepatoblastoma-targeted regimens		
PLADO	3	21
CDDP	2	0
C5V	0	2
C5VD	2	3
CarboDV	2	0
Soft-tissue sarcoma-derived regimens		
CEVAIE (+Cyclo)	5	0
ICE	5	0
VAC-IE	5	0
Other sarcoma regimens	5	4
Other regimens		
DV5Cyclo	1	1
DVCyclo	2	0
EAP	3	0
Other (non-standardized)	18	9

**Table 4 cancers-14-00272-t004:** Therapy and details of surgery.

	MRTL *n* = 55	SCUD *n* = 41	F-SCHB *n* = 22
Treated as HB	6 (11%)	40 (98%)	22 (100%)
Treated as MRTL	47 (85%)	0	0
Chemotherapy			
No oncological treatment	2 (4%)	1 (2%)	0
CHT with neoadjuvant intent	43 (78%)	33 (80%)	3 (14%)
Adjuvant chemotherapy	24 (44%)	20 (49%)	21 (95%)
PLADO regime (only)	1 (2%)	20 (49%)	15 (68%)
Soft tissue sarcoma regime			
No response	35 (64%)	29 (71%)	2 (9%)
Partial response	8 (15%)	9 (22%)	3 (14%)
Good response	7 (13%)	1 (2%)	16 (73%)
Data on response not available	5 (9%)	2 (5%)	1 (5%)
Surgery			
Upfront surgery	10 (18%)	8 (20%)	19 (86%)
Surgery after chemotherapy	16 (29%)	11 (27%)	2 (9%)
No surgical tumor resection	29 (%53)	22 (54%)	1 (5%)
Reoperations	2 (4%)	3 (7%)	2 (9%)
Liver resection	25 (45%)	18 (44%)	20 (91%)
Liver transplantation	1 (2%)	2 (5%)	1 (5%)
R-status (Biopsies excluded)			
R0	12 (22% of all, 46% of those with surgery)	13 (32% of all, 68% of those with surgery)	21 (95% of all, 100% of those with surgery)
R1	10 (18% of all, 38% of those with surgery)	5 (12% of all, 26% of those with surgery)	0
R2	4 (7% of all, 15% of those with surgery)	1 (2% of all, 5% of those with surgery)	0
Radiotherapy			
Adjuvant radiotherapy	3 (5%)	3 (7%)	0
Definitive radiotherapy	1 (2%)	0	0

**Table 5 cancers-14-00272-t005:** Outcome.

	MRTL *n* = 55	SCUD *n* = 41	F-SCHB *n* = 22
Mean/median Follow-up	35/24 months (SD 34)	37/18 months (SD 41)	85/84 months (SD 47)
Complete remission	14 (25%)	2 (5%)	21
Partial remission	10 (18%)	14 (34%)	0
Disease progression during initial therapy	31 (56%)	25 (61%)	1 (5%)
Relapse	11 (20%)	16 (39%)	10 (45%)
Local relapse	9 (16%)	7 (17%)	0
Distant relapse	5 (9%)	9 (22%)	10 (45%)
Mean time to relapse	5 months	4 months	10 months
DRD	42 (76%)	38 (93%)	3 (14%)
Alive with disease	2 (4%)	1 (2%)	0
Disease free at last follow-up	11 (20%)	2 (5%)	19 (86%)

## Data Availability

Publicly available datasets were analyzed in this study. The full search strategy used for this study is provided.

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
