# Peer review of "A Systematic Review and Meta-Analysis of Malignant Rhabdoid and Small Cell Undifferentiated Liver Tumors: A Rational for a Uniform Classification"

_cancers, 2022, doi:10.3390/cancers14020272_

Round 1
Reviewer 1 Report
This is an interesting and nicely written review article on a topic that remains problematic from diagnostic and therapeutic standpoints.
Recent articles and case series have recommended that SCUD should be considered an MRT based on its molecular characteristics (SMARCB1 deletion; and in rare cases SMARCA4 mutation), response to therapy (better response to sarcoma chemotherapy regimens than hepatoblastoma regimens) and outcome, instead of relying on morphology (absence or presence of rhabdoid morphology) for classification. As shown in Figures 4 and 5, majority of patients with SCUD are treated with HB chemo regimens that has proven to be mostly ineffective.
The authors used great methodology to review the currently published articles and gather data.
Minor edits and review of the English language are recommended (e.g. page 2, introduction section, 6th paragraph: INI1 is [remove is] positivity even in the small tumor cells was documented; page 7, section 3.2., 2nd paragraph: F-SCHB from SCUD und [change to and] MRTL and had and [remove and] acceptable RoB.)
Reviewer 2 Report
This is a very important review paper because it is the only review providing a systematic meta-analysis with respect to analyzing similarities and differences between patient clinical features, biological behaviour and treatment response in liver tumors classified as small cell undifferentiated hepatoblastoma (SCUD), malignant rhabdoid tumors of the liver (MRTL) and the separate unrelated entity hepatoblastoma with focal small cell histology (F-SCHB). This review provides a detailed review comparing the clinical features, treatment and outcomes of these entities, and shows convincingly that the two entities (SCUD and MRTL) clinically have similar presentations, and importantly do not respond to HB directed therapies, but instead respond best to sarcoma directed therapies, possibly with inclusion of radiation therapy. This is a very important point, since historically most of the cases reviewed as SCUD (65%) received HB directed therapy, whereas only 17% of MRLT received HB directed therapy, and HB directed therapy in the MRLT group may have been due to the fact that 36% of the MRTL cases were initially misdiagnosed. The third entity Hepatoblastomas with focal small cell features (F-SCHB) was shown to have a very different clinical presentation and response to treatment, similar to the presentation and response of expected for HB.
All summary clinical charts, treatment and outcome charts support these statements.
Very clearly written.
One typographic error - last word of last sentence of paper (replace SUCD with SCUD)
Reviewer 3 Report
This is an important and meaningful study on MRTL and SCUD, which are two of the rarest and most aggressive forms of pediatric liver cancer. This review is particularly useful for developing appropriate treatment strategies for patients with SCUD.
As mentioned in the text, most pediatric cancers have a high degree of certainty in histopathological diagnosis, so this study of a small number of patients is well worthwhile, despite the bias.
Also, this study has been adequately reviewed by taking on board the necessary processes.
However, you need to make sure that there is no misunderstanding regarding the CQ of the unified classification of MRTL and SCUD.
Major comments
If genetical analysis of SCUD is advanced, we can reach the same conclusion as in this study. However, now, we do not have enough information to consider MRTL and SCUD as genetically similar.
Although there are many cases of negative INI1 in both MRTL and SCUD, we speculate that genetic analysis is essential to put tumors that have been pathologically classified separately into the same category. Therefore, in this study, it is necessary to define MRTL and SCUD by another classification method, for example, clinical classification.
We recommend that the sentence "Results of this meta-analysis suggest that SCUD should be classified as a subtype of MRTL." in Key Insights and Messages be changed as follows.
For example, "Results of this meta-analysis suggest that MRTL and SCUD should be classified as the same category of therapeutic strategy.
I consider that this is more accurate and acceptable to many pediatric oncologists and pathologists.
